# Barriers to and enablers of quality improvement in primary health care in low- and middle-income countries: A systematic review

**Camlus Otieno Odhus**[1]*, **Ruth Razanajafy Kapanga**[2], **Elizabeth Oele**[3]

**1** Division of Health Research, Faculty of Health and Medicine, Lancaster University, Lancaster, United Kingdom, **2** County Department of Health, Kakamega County Government, Kakamega, Kenya, **3** County Department of Health, County Government of Kisumu, Kisumu, Kenya

\* codhus20@gmail.com

**Data Availability Statement:** All data extracted from primary research reports have been included with the submission as Supporting Information (S5 Table).

## Abstract

The quality of health care remains generally poor across primary health care settings, especially in low- and middle-income countries where tertiary care tends to take up much of the limited resources despite primary health care being the first (and often the only) point of contact with the health system for nearly 80 per cent of people in these countries. Evidence is needed on barriers and enablers of quality improvement initiatives. This systematic review sought to answer the question: What are the enablers of and barriers to quality improvement in primary health care in low- and middle-income countries? It adopted an integrative review approach with narrative evidence synthesis, which combined qualitative and mixed methods research studies systematically. Using a customized geographic search filter for LMICs developed by the Cochrane Collaboration, Scopus, Academic Search Ultimate, MEDLINE, CINAHL, PSYCHINFO, EMBASE, ProQuest Dissertations and Overton.io (a new database for LMIC literature) were searched in January and February 2023, as were selected websites and journals. 7,077 reports were retrieved. After removing duplicates, reviewers independently screened titles, abstracts and full texts, performed quality appraisal and data extraction, followed by analysis and synthesis. 50 reports from 47 studies were included, covering 52 LMIC settings. Six themes related to barriers and enablers of quality improvement were identified and organized using the model for understanding success in quality (MUSIQ) and the consolidated framework for implementation research (CFIR). These were: microsystem of quality improvement, intervention attributes, implementing organization and team, health systems support and capacity, external environment and structural factors, and execution. Decision makers, practitioners, funders, implementers, and other stakeholders can use the evidence from this systematic review to minimize barriers and amplify enablers to better the chances that quality improvement initiatives will be successful in resource-limited settings. PROSPERO registration: CRD42023395166.

**Funding:** The authors received no specific funding for this work.

**Competing interests:** The authors have declared that no competing interests exist.

## Introduction

As the world passes the halfway mark towards the sustainable development goals, many low- and middle-income countries (LMICs) are not on track to achieve universal health coverage (UHC) [1]. This is partly blamed on failure to ensure that health systems deliver high-quality care [2]. As many LMICs choose to strengthen primary health care (PHC)—upon which most people rely for everyday healthcare needs—in their efforts to accelerate the attainment of UHC, quality improvement (QI) initiatives related to PHC in LMICs require closer examination. This review synthesizes literature on barriers to and enablers of efforts by health workers and different stakeholders to improve the quality of PHC in LMICs.

The Agency for Healthcare Research and Quality (AHRQ) contends that "QI is essential to achieving the triple aim of improving the health of the population, enhancing patient experiences and outcomes, and reducing the per capita cost of care, and to improving provider experience" [3].

Improving the quality of PHC is fundamental to the achievement of health goals in LMICs [2]. PHC is all too important because up to eight in every ten people in LMICs depend on it to meet their health and care needs [1]. For the better part of modern healthcare history, the poor quality of healthcare has generated concerns among practitioners, researchers, and policy-makers [4–7] and those concerns have only grown louder [2]. This is because despite more, though increasingly uncertain, investments and rapid innovation, health outcomes have stagnated with rising inequalities in many LMICs [8, 9] potentially leading to wastage, harm, and even preventable deaths [10].

While barriers (constraints or limitations) prevent the realization of full benefits from quality improvement (QI) interventions, enablers (also known as promoters, facilitators, or motivators) unlock the potential of such interventions and typically enhance the desired level of quality of PHC. Both range from the individual or micro (e.g., nurse manager knowledge and behaviour), to institutional-organizational or meso (e.g., shared beliefs, attitudes and practices at a health centre or hospital), to system-wide and societal or macro influences, e.g., implicit, or explicit values that drive QI culture, priority-setting, or investments.

First, it is necessary to define key terms. PHC is challenging to define because it includes or precludes different packages of health services in different contexts. Perhaps it is due to this challenge that the World Bank, the World Health Organization and others [1] opted to define PHC rather broadly as "a health- and social-service delivery platform or system uniquely designed to meet communities' health and health care needs across a comprehensive spectrum of services—including health services from promotive to palliative—in a continuous, integrated, and people-centred manner." PHC services are often attuned to the prevailing socio-economic, political and historical contexts of communities, in addition to the financial and health workforce considerations in the given country setting [11, 12].

Competing but comparable definitions of quality of care which hold important implications for how QI (in healthcare) is defined and operationalized have been proposed by the World Health Organization [13], by the United States National Academy of Medicine, formerly Institute of Medicine or IOM [6], and others [14, 15]. However, consensus remains elusive [14]. But, QI—with roots in manufacturing in 1920s—can be defined as a framework with tools, approaches, techniques, and skillsets including assessment and measurement, goal-setting, and shifts in mindsets geared towards improving equity, access, effectiveness, patient-centeredness, and safety of healthcare [15, 16]. Ongoing debates on the level (individual or population), scope (bounded setting or whole systems), and approaches (evidence-based practice, multidisciplinary) to healthcare QI are unlikely to be concluded soon [17–19]. The review

considered the lack of consensus by being as inclusive as possible, avoiding a one-size-fits-all approach to defining QI.

Some of the existing reviews have synthesized evidence on patient safety culture in Latin American Hospitals [20], barriers and enablers to the provision of emergency obstetric care in Nigeria [21] and in LMICs [22], and interventions to improve anti-retroviral therapy programmes in sub-Saharan Africa (SSA) [23]. A COCHRANE review studied the use of reminders in health care [24]. Notably, an umbrella review [25] describing the influence of contextual factors on hospital QI using the Model for Understanding Success in Quality (MUSIQ) tool [26] found that previous systematic reviews overwhelmingly included studies from high income countries in North America, Europe, and Southeast Asia and very few from LMICs (Egypt, South Africa, Zambia, Sudan, Costa Rica, Brazil, and Argentina). A more recent realist-inspired review [27] confined itself to a specific type of QI, namely "QI collaboratives" to investigate contexts, mechanisms and outcomes but still included only five (out of 32) primary studies from LMICs. Still, other reviews have confined themselves to 'training and measurement' [28–30] and patient safety education [31, 32]. No systematic review was found that synthesized literature from LMICs to inform holistic QI policy and practice specifically in PHC.

The systematic review aimed to describe the barriers to and enablers of QI within PHC in LMICs. The review sought to answer the following three closely related questions:

1. What are the barriers to and enablers of QI in PHC in LMICs?

2. What is the shared knowledge, beliefs, values, attitudes, and practices (collectively called 'culture') of LMICs' health workers and stakeholders regarding QI in PHC?

3. What micro (individual or personal), meso (institutional or organizational) and macro (societal or structural) factors motivate health workers and managers involved in PHC QI in LMICs?

## Materials and methods

### Review approach

An integrative approach [33] incorporating narrative synthesis [34] for this systematic literature review. Integrative reviews are suitable for combining studies from disparate methodological approaches such as mixed methods and qualitative studies explicitly and has played an expanding role in health systems and policy research [33], contributing to evidence-based policy and practice. The framework for integrative review commences with problem identification, proceeds through a literature search, appraisal of data and analysis, before concluding with data presentation.

A narrative approach to evidence synthesis relies on 'storytelling', as its name suggests, and is commensurate with the overall integrative review approach [34]. Correctly performed, narrative synthesis can minimize bias in reviews, ensuring that the eventual review output can be trusted by policymakers and practitioners alike. In the present review, this approach was used to enrich the data analysis and presentation stages of the integrative review. The findings of this systematic review incorporating primary studies on different aspects of QI were synthesized narratively. To comprehensively answer the review question, both mixed methods and qualitative studies investigating barriers, enablers, culture, and other contextual influences on diverse QI interventions in PHC in LMICs were included.

## Literature search strategy

The search for primary research reports was performed in January and February 2023. Electronic databases (MEDLINE, PSYCHINFO, EMBASE and CINHAL) were searched using a mix of free-text (key words in S1 Table) and Medical Subject Headings (MeSH terms), refined for each database using EBSCO interface. More search explored TRIP, Academic Search Ultimate, Web of Science, Scopus, and Africa Index Medicus. The key terms used to develop the literature search strategy drew upon the SPIDER mnemonic [35] included "Quality Improvement" AND "Primary Health Care" AND "Low- middle-income countries".

A scoping search was first used to check how studies are indexed and the relevant key words and synonyms. It was also used to test and refine the search strategy. *A priori* search strategy was then developed and applied to each database flexibly. A sample search strategy used for MEDLINE is contained in S1 and S2 Figs for Proquest. Neither time nor language filters were applied at this stage. Boolean and near field operators were used to expand and narrow the search as appropriate. A geographic search filter for LMICs developed by the Cochrane Collaboration's Effective Practice and Organisation of Care (EPOC) group [36] was applied to exclude high income countries. Literature was searched and retrieved in January and February 2023.

Grey literature including dissertations and thesis reports were sought from PROQUEST and the WHO and UNICEF public websites were also searched as was the preprint server, Medrxiv. To further reduce publication bias, Overton.io (an open research initiative to expand access to grey literature from LMICs) was also searched for grey literature. Finally, selected QI-focused journals (Health Policy and Planning, Implementation Science, International Journal for Healthcare Quality, BMJ Open Quality, Journal for Healthcare Quality, BMJ Quality and Safety, Journal of Health Services Research) were hand-searched as were reference lists of systematic reviews in the field of QI.

## Study selection

All (n = 7,077) reports were imported into Rayyan systematic review management (web platform) where (n = 4,110) duplicates were removed automatically and manually. Each title and abstract (n = 2,967) was screened independently by at least two reviewers and included (n = 227) if they were deemed relevant. Conflicts throughout the selection process were resolved by consensus. At full text review, reports were read multiple times and subjected to inclusion and exclusion criteria as shown in Table 1. Inclusion and exclusion criteria were derived from the SPIDER mnemonic [35] and signified the information power of the primary research report to contribute answers to the review question(s).

Eventually, 50 research reports from 47 studies were found that met the inclusion criteria for the systematic review following independent decisions by reviewers. Fig 1 is a PRISMA flow chart showing results of the study selection process [37].

## Assessment of study quality and relevance

The Mixed Methods Appraisal Tool, MMAT, checklist [38] was used to critically assess the quality of all 50 included full text reports prior to data extraction. MMAT checklist was especially suitable because it was developed for systematic reviews incorporating primary studies from different designs (qualitative, quantitative, and mixed methods). The first two screening questions ask whether there were clear research questions and if the data collected allowed the primary researchers to address the study's research question. For qualitative studies the tool has five themes (with yes, no, or can't tell response options): coherence between methodology and research question, coherence between data collection methods and research question,

**Table 1. Inclusion and exclusion criteria.**

| SPIDER element | Include | Exclude |
|---|---|---|
| Sample | Facility-based HCWs<br>Community-based health workers<br>Health managers, policymakers and stakeholders across the health system | Exclude if others included and lumped alongside these in findings. |
| Phenomenon of interest | Quality improvement (not just quality of care or general health systems capacity or situation assessment)<br>Quality includes safety, effectiveness, patient-centeredness, timeliness, efficiency, equity in health uptake/access, utilization, or outcomes.<br>Must be primary care or primary health care oriented, reported separately from tertiary and referral level. | Exclude high income country context, exclude health technology assessments, exclude other systematic and umbrella reviews. Also exclude very low-quality studies (judged by consensus) and those from tertiary care (university/teaching and research hospitals). Exclude editorials and opinion pieces, economic evaluations, and clinical case reports.<br>Include only primary empirical research (mixed or qualitative) reporting enablers to and barriers of quality improvement from perspective of health workers, health managers or regulators as study participants. |
| Design | Mixed methods and Qualitative designs.<br>Mixed methods papers have qualitative data detailing enablers or barriers. | Quantitative design with no discernible data on contextual drivers of QI measured or reported |
| Evaluation | Intervention to improve quality of health care i.e., efforts introduced to Change quality from level X to Y or measured from time X to time Y i.e., a QI initiative rather than just a measurement of quality of care. | Economic evaluations with no accompanying contextual data<br>One-off measurement seeking to perceptions of stakeholders on quality of care rather than on QI intervention/initiative/ project |
| Research type | Qualitative data reported separate from quantitative findings in mixed methods.<br>Qualitative research findings qualitatively reported (not quantified in percentages or numerical values).<br>Semi-structured or in-depth Interviews, focus groups, observation, ethnography etc. | Surveys, Randomized Trials with no process evaluations reporting barriers or enablers of QI initiative or QI project |

adequacy of findings given the data, coherence between the interpretation of findings and the data, and coherence in the research cycle from data sources, collection, analysis, and interpretation. To assess mixed methods studies MMAT focuses more on the appropriateness of mixing methods, whether the various methodologies were suitably combined, and how rigour and trustworthiness for each research tradition was maintained in the primary research. Each quality criteria entails "yes", "no", or "can't tell" response options. In keeping with best practice for integrative reviews and narrative synthesis, no quantitative scoring was done, and no study was excluded from the analysis based on the results of the critical appraisal, but the strengths and limitations of each study were considered in the ensuing synthesis.

## Data extraction

The corresponding author extracted data from all 50 included reports while each of the other co-authors independently extracted data from a smaller sample of 23, about half of all included reports. A fourth reviewer extracted data from one report. A comparison of critical appraisal and extracted datasets showed no major inconsistencies. The bespoke data extraction form also had sections to capture QI theory (of change), description of the QI intervention, study setting, sample and population, barriers, and enablers as well as motivations and other contextual influences. Lastly, data on study conclusions, limitations and strengths, and recommendations (where available) were included. Data extraction made use of Microsoft Office Forms, hosted online.

## Data analysis and synthesis

Data analysis involved the use of two frameworks commonly applied in QI research. The MUSIQ model developed by Kaplan et al. [26] was predominantly used, complemented with

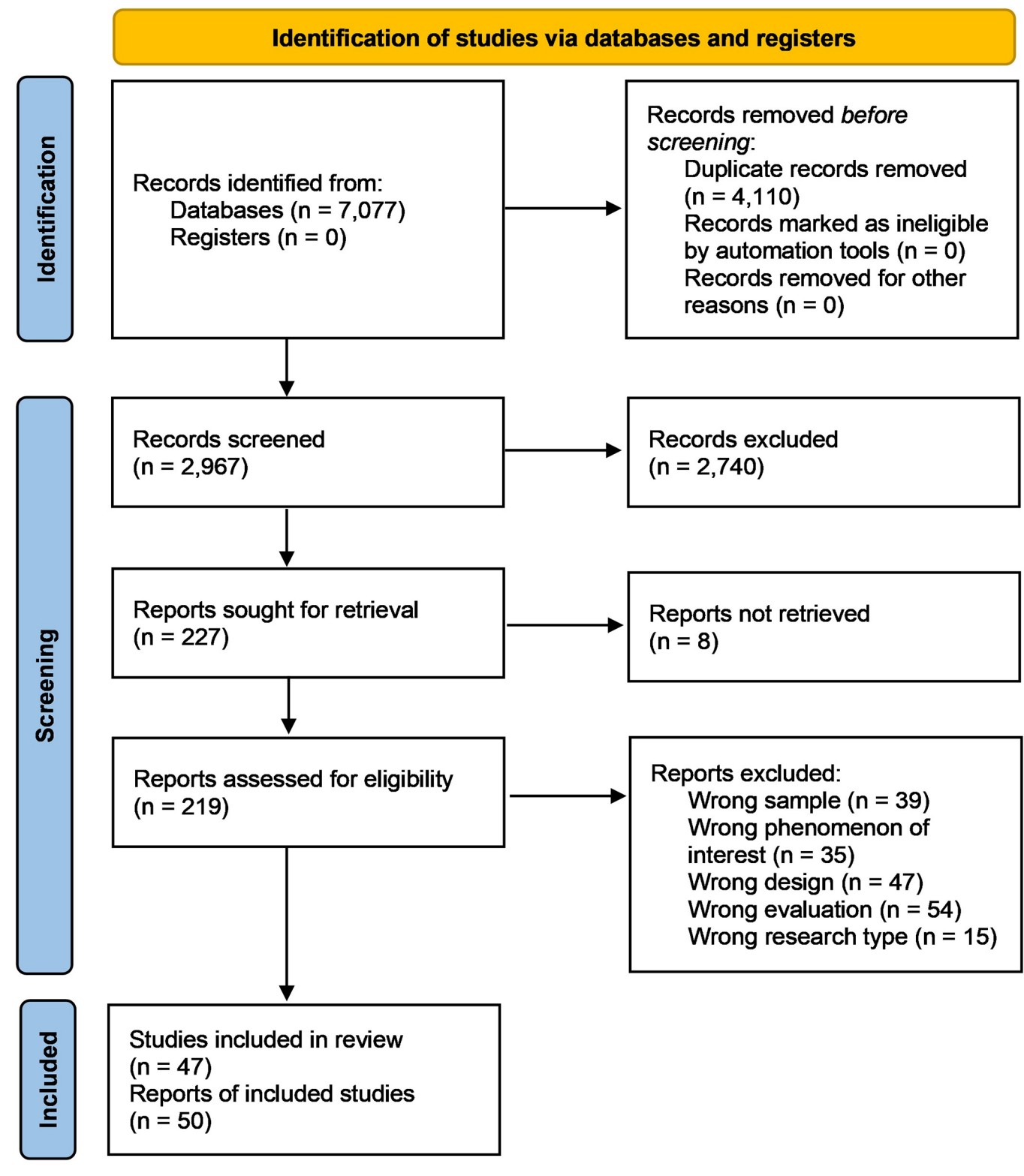

**Fig 1. PRISMA flow chart.**

the Consolidated Framework for Implementation Research, CFIR, [39, 40]. MUSIQ contains concepts for understanding the external context of QI interventions while CFIR complemented this by providing a way to organize attributes intrinsic to the QI intervention itself. Concepts from these two frameworks were deductively applied to the entire dataset of 50 research articles and inductive coding with labels grounded in the data was done where data did not fit into the *a priori* coding framework. The entire process was iterative with multiple revisions. Atlas.ti version 9 (ATLAS.ti Scientific Software Development GmbH, Berlin) was used for coding and categorization.

First, to enable detailed and systematic analysis of this large dataset and in concert with the integrative review approach, studies were classified and grouped by geographic region, country income status and study topical focus. This allowed systematic comparison of studies and integration of their findings. Next, deductive codes from MUSIQ and CFIR were applied to the data extracted from studies in addition to new (inductive) codes. Codes were then grouped into categories (still informed by MUSIQ and CFIR) before being displayed in tables and matrices and network diagrams. Through comparisons and contrasts, noting surprising or unique findings and variability within and across subgroups, the analysis moved into the final phase. Here, a description of patterns in the form of themes concluded the analysis by narratively synthesizing subgroup patterns into an overall picture to address the review's three aims: to describe the evidence on barriers and enablers of PHC QI; to uncover individual motivations (of health workers, managers and other stakeholders) for undertaking QI; and to describe what the culture (shared knowledge, language, or artifacts) of QI looks like in PHC settings in LMICs contexts. Verification of results was done by going back to primary studies to ascertain the link with eventual conclusions (S2 Table).

## Results

### Characteristics of included studies

Fifty primary research reports were included in the analysis. Twenty-eight had mixed methods design while 22 were qualitative, as summarized in Table 2. Signifying increasing interest in PHC QI by researchers, 41 of the studies were published in the last five years (2018 to date) while only nine were reported between 2012 and 2017. Although the review had aimed to include studies since 2000, none of the included studies covered the period 2000 to 2011. Turning to geographical coverage (S2 Table) for this systematic review that sought literature from LMICs on barriers and enablers of QI in relation to PHC, forty-one of the studies were based in sub-Saharan Africa (SSA), seven from Asia and two from Latin America (Costa Rica which is upper middle-income and Haiti which is lower middle-income). All seven studies based in Asian countries came from lower middle-income settings (India had three while Indonesia, Tajikistan, Papua New Guinea, and Sri Lanka had one study each). Out of the forty-one studies from SSA, 37 reported research conducted in a single-country set-up while four covered multiple countries. In total, research reports covered 45 SSA countries. A close examination revealed that three publications [41–43] were likely from the same QI intervention in Tanzania and Uganda and a further two publications [44, 45] were from the same project in Nigeria.

### Topics from included research

Researchers overwhelmingly focused on topics related to improvements in maternal and child health (MCH) with twenty-nine studies, including some two conducted in Kenya [46, 79] and some five that focused on maternal and perinatal deaths: in Ethiopia [63]; in Benin [89]; in South Africa [65]; in Rwanda [64] and in Rwanda, Tanzania, Zimbabwe and Nigeria [62]. A

**Table 2. Characteristics of included studies.**

| Author | Country/ setting | Topic | Purpose/ Aim (as described in the study) | Research design |
|---|---|---|---|---|
| Baker et al. [41] | Southern Tanzania: Tandahimba district | Understanding QI from perspective of health workers | To investigate how different components of a collaborative QI intervention were understood and experienced by health workers, and therefore contributed positively to its mechanisms of effect. | Qualitative process evaluation with semi-structured interviews. |
| Tancred et al. [42] | Sothern Tanzania: Tandahimba district Uganda: Mayuge district | Community maternal newborn child health | We describe the experience implementing EQUIP's QI approach at the community level for increased demand for maternal and newborn health services and improved community-level maternal and newborn care practices. | Qualitative data as part of in-depth mixed methods process evaluation. |
| Tancred et al. [43] | Southern Tanzania: Tandahimba district | Maternal and newborn health at community level | To understand the perceptions and motivations for the behaviours of both those engaged in implementing quality improvement and those affected by their problem-solving strategies. | A mixed methods process evaluation. |
| Eboreime at al. [44] | Nigeria: Kaduna state | Decentralized primary health care planning | To evaluate the effectiveness of DIVA as a model for improving health system performance through integrated PHC operational planning in Kaduna, Nigeria. | Embedded mixed methods evaluation. |
| Eboreime et al. [45] | Nigeria: Kaduna state | Decentralized primary health care planning | To explore the role of actors and context in the implementation and sustainability of diagnose-intervene-verify-adjust (DIVA) by comparing experiences between Nigerian local government areas (LGAs) (analogues of districts) in Kaduna state. | An integrated mixed methods approach. |
| Giessler et al. [46] | Kenya: Four government health facilities in Nairobi and Kiambu Counties | Maternal health (patient centred care) | Study focuses on the experiences of both clinical and non-clinical staff who took part in a quality improvement collaborative focused on improving patient centred care for maternal health and family planning in public facilities in Kenya. | Descriptive qualitative exploration using semi-structured interviews. |
| Odusola et al. [47] | Nigeria: Kwara State | Hypertension prevention and care using health insurance | To explore perspectives of insurance managers and primary care staff on factors that might inhibit or facilitate the implementation of high-quality hypertension care in practice. | Qualitative design and semi-structured individual interviews. |
| Pesec et al. [48] | Costa Rica: nationwide | Health care reforms: collection and use of data for quality improvement | To identify the sources of PHC data in Costa Rica's healthcare system and describe how these data are used for quality improvement. | Qualitative methodology with in-depth, in-person semi-structured interviews. |
| Lall et al. [49] | South India: Kolar, Karnataka State, in three government healthcare facilities | Non-communicable diseases: service reorganization | We critically analyse the implementation process using implementation and quality improvement frameworks to identify contextual factors that may have resulted in the differential uptake of interventions at the different primary health care centres. | Mixed methods: Case experimental design with observation and the implementation of interventions. |

*(Continued)*

**Table 2.** (Continued)

| Author | Country/ setting | Topic | Purpose/ Aim (as described in the study) | Research design |
|---|---|---|---|---|
| Wakida et al. [50] | Uganda: Mbarara district, about 270 Kilometers by road, southwest of Kampala | Clinical practice guidelines (CPG) implementation: mental health disorders | This study aimed to assess the feasibility and acceptability of an educational intervention towards improvement of the primary health care practitioners' uptake of the clinical practice guidelines in integrating mental health services into primary health care in Mbarara district, southwestern Uganda. | Descriptive cross-sectional qualitative study. |
| Bogren et al. [51] | Democratic Republic of Congo: South Kivu Province | Maternal and newborn health: health worker training | To explore contextual factors influencing a training intervention focusing on health care practice during childbirth. | Qualitative research design, and data was collected through focus-group discussions (FGDs). |
| Tibeihaho et al. [52] | Uganda: 13 districts | Institutionalizing continuous quality improvement | To understand how the continuous quality improvement processes introduced by the CODES project were institutionalized at the district level. | Qualitative research design: District documents relevant to the continuous quality improvement process were also reviewed. |
| Gage et al. [53] | Zimbabwe: Centenary, Chipinge, Mwenezi, Binga and Mangwe districts | Continuous quality improvement through performance-based financing | To evaluate the continuous quality improvement (CQI) pilot in Zimbabwe: first, what is the effect of the CQI model on quality of care and second, what factors enabled or impeded quality improvements during CQI implementation? | Mixed methods approach: quantitative analyses of the PBF quality checklists using quasi-experimental design. And qualitative analyses of document reviews, in-depth interviews, and focus group discussions (FGD). |
| Tiruneh et al. [54] | Ethiopia: Selected rural areas | Maternal newborn health | To evaluate the effect of the PC-Solutions strategy on improving MNH care behaviours and practices in selected rural areas of Ethiopia. | Mixed-methods research. We used before-and-after cross-sectional survey. The qualitative method included. |
| Patterson et al. [55] | Malawi: facilities that provided basic or comprehensive childbirth services. | Quality of care and culture | To identify what would be necessary to foster organizational cultures in Malawi closer to the hypothetical "culture of quality" outlined in the public health literature. | Ethnographic data were generated through observation and semi-structured interviews. |
| Demes et al. [56] | Haiti: Northern Department | A fingerprint initiative to curb absenteeism | To explore the quality improvement initiatives in the context of Haiti by assessing the process and outcomes of the implementation of the fingerprint initiative in three health facilities in the Northern Department. | Exploratory and qualitative descriptive study. |
| Kim et al. [57] | Uganda: Busia and Oyam districts | Quality improvement collaborative for community-based family planning | To identify the factors that were supportive of the community-based quality improvement collaborative implementation, as perceived by the collaborative actors and in relation to the Bruce Framework. | Descriptive mixed methods process evaluation design: desk review of program documents, extraction of program monitoring data, and qualitative research methods. |
| Lokossou et al. [58] | Benin: Savè-Ouèssè (SAO) health zone | Community health workers: motivation, retention, and performance | To present the results of implementing quality improvement approach at the community level in the Savè-Ouèssè (SAO) health zone in Benin and to examine the perceptions of the actors involved in the implementation to strengthen the local components of health systems. | Mixed-methods approach that included a quantitative (analysis of indicator trends) and a qualitative study. |
| Vail et al. [59] | India: Bihar state | Newborn resuscitation | To characterize the logistical, cultural, and structural barriers to the use of evidence-based practices in immediate neonatal care, defined as care required during the immediate transition to post-natal life, and Neonatal resuscitation. | Qualitative using semi-structured interviews. |

(*Continued*)

**Table 2.** (Continued)

| Author | Country/ setting | Topic | Purpose/ Aim (as described in the study) | Research design |
|---|---|---|---|---|
| Visser et al. [60] | South Africa: Greater Tazneen sub-district (municipality) of Limpopo province | HIV/AIDS care and treatment: nurse-monitored care (task shifting) | To evaluate the quality of care provided at three selected nurse-initiated and managed anti-retroviral therapy facilities in the Greater Tazneen sub-district of Limpopo province and, to explore the effects of clinical mentoring and support on improving the quality of care. | A mixed methods study that used concurrent quantitative and qualitative research methods was conducted. |
| Jaribu et al. [61] | Southern Tanzania, Ruangwa district, located in Lindi Region | Institutional childbirth services | We used in-depth interviews with health workers at various levels in the health system to explore their perception of the QI intervention and to identify facilitators and barriers in relation to QI implementation. | Qualitative study with in-depth interviews. |
| Kinney et al. [62] | Four sub-Saharan African countries: Rwanda, Tanzania, Zimbabwe, Nigeria | Maternal and perinatal death surveillance and response | The aim of this study was to systematically assess the level of implementation of maternal and perinatal death surveillance and response (MPDSR) in four sub-Saharan African countries, applying a standardised scoring methodology, and to describe common facilitators and barriers to sustainable MPDSR practice. | Mixed methods: qualitative and quantitative data collection methods —observations, review of documents and semi structured key informant interviews. |
| Ayele et al. [63] | Northern Ethiopia: Tigray region | Maternal and perinatal death surveillance and response | To assess the implementation status of MPDSR and its associated factors as well as explore the barriers and facilitators of MPDSR implementation and operation in Tigray region, Northern Ethiopia. | Mixed methods: quantitative (facility-based cross-sectional study) and qualitative (in-depth interviews and focus group discussions) approaches. |
| Tayebwa et al. [64] | Rwanda | Maternal and perinatal death surveillance and response | To assess experiences in implementing maternal and perinatal death review, and/or integrated MPDSR processes in Rwanda by identifying factors that have affected its implementation | Mixed methods with qualitative and quantitative data. |
| Kinney et al. [65] | South Africa: Western Cape | Perinatal death audit programme | To understand the 'how' or 'why' of sustained implementation, allowing for comparison across settings to gain insights on factors influencing sustained implementation of perinatal audit. | Multiple Case study. |
| Basenero et al. [66] | Namibia: three regions with high burdens of HIV—Khomas, Ohangwena, and Zambezi | Integrating Hypertension and HIV/AIDS care | In this work, we report the implementation of a quality improvement collaborative—the Namibia Project for Retention of Patients on ART (NAMPROPA)—whose objective was to improve uptake of HTN screening and treatment in routine HIV care in Namibia. | Mixed methods. |
| Schuele & MacDougall [67] | Papua New Guinea: Madang and Morobe Provinces | Accreditation of lower-level health facilities to higher level facilities | To critically examine driving and restraining forces in the implementation process of the national health service standards, understand how hidden power relations work in the implementation process, and assess agenda setting to influence change. | Qualitative with semi-structured interviews and FGDs. |

(*Continued*)

**Table 2.** (*Continued*)

| Author | Country/ setting | Topic | Purpose/ Aim (as described in the study) | Research design |
|---|---|---|---|---|
| Hutchinson et al. [68] | Uganda: Kayunga District | Malaria surveillance | The aims were: (i) to describe the context in which, and the processes through which, the collaborative improvement (CI) intervention effected change; (ii) to identify any factors that support or undermine CI; and (iii) to investigate for any unintended consequences of the CI intervention. | Qualitative study. |
| Yapa et al. [69] | South Africa: Hlabisa sub-district of KwaZulu-Natal, 220 km north of Durban | Antenatal HIV Care and Testing | To identify determinants of practice, and whether 'normalisation' of continuous quality improvement (CQI) into routine services could occur in this setting, by examining the following: (i) health worker participation in CQI by describing 'dose' and 'reach'; (ii) the 'black box' of implemented changes in practice; (iii) time trends in endpoint achievements and time to intervention uptake; and (iv) CQI mentor and health worker experiences of implementing the intervention. | Convergent mixed methods: Process evaluation of CQI as implemented in our stepped-wedge cluster RCT. |
| Limato et al. [70] | Indonesia: 3 Puskesmas in Cianjur district, West Java province | Primary health care quality improvement | This study aimed to contribute to improving health service quality in the primary health care system in Indonesia. | Qualitative: in-depth interviews. |
| Umunyana at al. [71] | Rwanda | Management of birth asphyxia | Our study aimed to show that a capacity development package focused on mentorship as part of a larger quality improvement strategy would contribute to improved clinical skills and better neonatal outcomes for birth asphyxia at scale. | Mixed methods before-after design. |
| Stover et al. [72] | Ethiopia: Amhara and Oromiya Regional Health Bureaus | Maternal Newborn health (district level improvement) | This article describes the methods by which and the extent to which Maternal and Newborn Health in Ethiopia Partnership was able to develop the capacity of coaches and teams to support continuous improvement in CMNH care. | Mixed methods: Surveys and individual interviews |
| Chandani et al. [73] | Malawi and Rwanda | Supply chain systems for CHW child health commodities | This paper will discuss the results of scaling proven, simple demand-based resupply procedures, using mobile technology and traditional methods for communication, and establishing multilevel, performance-driven QI teams in Malawi and Rwanda, and the potential contributions these interventions had on supply chain outcomes for CHWs. | A mixed-method approach; qualitative data was collected using a case study methodology, and quantitative data was collected. |
| Horwood et al. [74] | South Africa: KwaZulu-Natal province | Electronic clinical decision-making support systems (CDSSs): electronic integrated management of childhood illnesses (eIMCI) | To track eIMCI uptake and prospectively explore their experiences of eIMCI implementation in primary health care (PHC) clinics in one district in Kwa Zulu Natal. | Longitudinal mixed methods study, which was nested within a randomized controlled trial (RCT). |

**Table 2.** (*Continued*)

| Author | Country/ setting | Topic | Purpose/ Aim (as described in the study) | Research design |
|---|---|---|---|---|
| Mantell et al. [75] | South Africa: The City of Tshwane, Gauteng Province, and Bojanala in Northwest Province | Ward-based primary healthcare outreach teams | This paper examines program implementation and barriers and successes from the perspectives of the national department of health, implementing partners, facility-level staff, and the outreach team. | The process evaluation used a parallel convergent mixed-methods design, with concurrent collection of qualitative and quantitative data at multiple levels. |
| Thekkur et al. [76] | Sri Lanka: nine provinces | Primary Healthcare System-Strengthening | To assess if primary medical care institutions were re-organised according to the standards endorsed by the ministry of health, and to explore the challenges perceived by the healthcare workers implementing this project | An explanatory mixed-methods study with quantitative component (cross-sectional descriptive study) and a qualitative component. |
| Mutambo et al. [77] | South Africa: KwaZulu-Natal Province | Child-friendly spaces (child-centred HIV care) | To explore the experiences of health care workers, primary care givers and HIV seropositive children on the use of child-friendly spaces in PHC facilities in KwaZulu-Natal | Qualitative explorative, descriptive, and contextual design. |
| Schierhout et al. [78] | India: West Godavari District in rural Andhra Pradesh state | Digital health interventions and cardiovascular disease | This study aims to identify variation in outcomes and implementation of SMARTHealth India, a cluster randomised trial of an ASHA-managed digitally enabled primary health care (PHC) service strengthening strategy for cardiovascular disease risk management, and to explain how and in what contexts the intervention was effective. | Realist evaluation and an explanatory sequential mixed method. |
| Djellouli et al. [79] | Burkina Faso: Kaya district Kenya: Kwale County (Matuga constituency) Malawi: Ntchisi district Mozambique: Chiuta district | Maternal and Child Health—post natal care | This evaluation aimed to uncover how the interventions implemented resulted in increased uptake, frequency of delivery and quality of evidence based postpartum care and what worked, for whom and within which contexts. | Case study design and realist evaluation methods using mixed methods. |
| Werner et al. [80] | Tajikistan | Business Plans (health facility management tools) | The objectives of this study are (i) to describe the history, process of implementation and consolidation of Business Plans in the Tajik health system by means of the ExpandNet/WHO framework, (ii) to identify barriers and facilitators to scaling up and based on that (iii) to extract lessons learnt related to scaling up health innovations. | Qualitative. |
| Coulibaly et al. [81] | Mali: 3 of the 10 Health Districts in Koulikoro region | Performance-based financing | How is performance-based financing implemented and adapted to the socio-political, health and institutional contexts in Mali? | Qualitative multiple case study approach. |
| Bradley et al. [82] | Ethiopia: 4 regions | Rural primary health care | To generate hypotheses about factors that may explain the variation in performance across primary health care units. | An in-depth qualitative study, drawn from a longitudinal study |
| Sukums et al. [83] | Tanzania: Lindi rural district Ghana: Kassena-Nankana district | Antenatal/ intrapartum care and performance-based incentives | To describe health workers' acceptance and use of the electronic clinical decision support system for maternal care in rural PHC facilities of Ghana and Tanzania and to identify factors affecting successful adoption of such a system. | Longitudinal mixed methods study. |

(*Continued*)

**Table 2.** (Continued)

| Author | Country/ setting | Topic | Purpose/ Aim (as described in the study) | Research design |
|---|---|---|---|---|
| Nahimana et al. [84] | Rwanda: Kirehe and South Kayonza districts in the Eastern Province | Newborn care | To describe the integration of key elements of All Babies Count (ABC) program into routine systems and the results evaluating 12 months sustainability of improvements seen during the ABC program and factors related to the success and challenges of sustainability. | Mixed methods convergent sequential design. Quantitative evaluation using a pre-post design. Focus group discussions and in-depth interviews. |
| Quaife et al. [85] | Ethiopia: 7 intervention districts matched with 7 comparison districts (*woredas*) | Health worker knowledge and motivation | To evaluate whether and how the Ethiopia Health Care Quality Initiative affected health worker knowledge and motivation, and if effects differed by cadre. | We used mixed methods, combining a repeated quantitative survey with supporting in-depth qualitative interviews. |
| Olaniran et al. [86] | Nigeria: Lagos health system | Maternal and neonatal health and patient experience and satisfaction | To contribute to the evidence base about how and why QI works using the implementation of the national healthcare quality improvement and how this was adapted in the Lagos health system. | A qualitative study using a multiple-case study design. Combined an exploratory with an explanatory approach. |
| Manzi et al. [87] | Rwanda: Kirehe and Southern Kayonza districts | Child health (mentorship) | To inform program implementers and policy makers of the key components needed and potential barriers and resistance which can be addressed proactively when implementing similar health facility-based mentorship interventions. | A qualitative study using focus group discussions (FGDs) and in-depth interviews. |
| Werdenberg et al. [88] | Rwanda: Kirehe and Southern Kayonza districts | Newborn health | This paper reviews the implementation process and implementation outcomes of the ABC initiative including feasibility and fidelity, acceptability, self-reported changes in health care worker (HCW) attitudes and practice of QI, implementation and the resulting change package. | Mixed methods: quantitative surveys, and qualitative data from FGDs and review of program documents. |
| Hounsou et al. [89] | Benin | Maternal and perinatal survival | The present study aims to examine whether, and to what extent, implementation of the four components of MPDSR took place in Benin and identify lessons for improving MPDSR implementation going forward | Retrospective, mixed-methods study. |
| Pallangyo et al. [90] | Tanzania: Dar es Salaam city area | Maternal and child health (postpartum care) | To explore the strategies used by facilitators and health care providers within a facilitation intervention to improve post-partum care in government-owned health institutions in Ilala suburb in Dar es Salaam, Tanzania. | A qualitative design with focus group discussions (FGDs) and intervention documentation. |

summary of studies by topic of focus is contained in supplementary files (S3 Table). As well, five studies documented QI in relation to non-communicable diseases: hypertension service coverage in Nigeria [47]; digital health interventions for cardiovascular health in India [49, 78]; mental health services in Uganda [50]; and integration of hypertension and HIV services in Namibia [66]. Three studies explored QI in HIV/AIDS: nurse-monitored HIV/AIDS care and treatment as part of task-shifting [60] and antenatal HIV care and testing [77] in South

Africa, and service expansion through integration in Namibia [66]. One study [68] sought to further the understanding of collaborative QI in malaria surveillance in Uganda. Three studies: in Rwanda and Nigeria [73]; in South Africa [74]; and in India [78] investigated the application of digital interventions to improve PHC service delivery.

## Themes

Barriers to and enablers of QI in PHC at micro, meso- and macro- level were distilled into six themes (Fig 2), guided by the MUSIQ model and the CFIR, and are described next. Themes are closely related and mutually interacting (see also S4 Table).

**Theme 1: Microsystem and individual health worker(s) motivation.** The willingness and commitment of individual health workers to make improvements, their ability and self-efficacy regarding change efforts, shared values, beliefs, and norms that affect teamwork, interpersonal communication and decision making, and the capacity of health workers and managers to lead QI can constrain or promote QI in PHC settings. Three multi-country studies in Sub-Saharan Africa [62, 79, 83], 19 single country studies in SSA (three apiece in Nigeria, South Africa and Uganda, two each in Benin, Ethiopia and Rwanda, and one study across Benin, Kenya, Mali, Tanzania and Zimbabwe reported various aspects of individual- and microsystem- level barriers to and enablers of QI along with the two studies [48, 56] from Latin America (Costa Rica and Haiti). Studies in Indonesia [70]; in Sri Lanka [76]; in India [49]; in Tajikistan [80]; and in Papua New Guinea [67] also discussed aspects of microsystems and individual health worker motivations for QI. All studies had good quality ratings using the mixed methods appraisal tool.

Health workers and other PHC stakeholders reported that job satisfaction arising from participating in QI activities was an important source of motivation, encouraging them to increase

| Microsystem and health workers | Commitment, willingness, ability and self-efficacy of facility-based workers; shared values, beliefs and norms that affect teamwork, decision making, and interpersonal communication. |
|---|---|
| QI intervention attributes | Evidence underpinning QI initiative, trialability, ease of integration, participants' perception of benefit and involvement in its design, costs, potential for scale up, and perceived sustainability of QI initiative. |
| Implementing organization and QI team | Buy-in (ownership), organizational culture and maturity on quality, leadership, and decision-making, tenure, prior experiences, cohesion, and skills of the implementing team; involvement of seniour (specialist) doctors/nurses. |
| Health systems support and capacity | Availability, adequacy, and distribution of resources, capacity (staffing, supplies, equipment, physical space, infrastructure, data, learning and knowledge systems, management of patient referrals, and leadership and governance). |
| External environment and structural factors | Larger context of QI: the social, geographic, economic, political, legal, and other normative aspects that shape societal and national health systems priorities has indirectly or directly effects. |
| Implementation of QI | Includes dosage (intensity) and reach (coverage), and how the QI intervention is executed (scope, quality, consistency, time, and other inputs) towards intended results. |

**Fig 2. Summary of themes.**

efforts and stirring up their desire to address the community's health needs. Added to this, health workers felt extrinsically motivated by financial and non-financial incentives as was the case in Nigeria [46] where Odusola and colleagues found that such inputs bolstered efforts to expand hypertension preventive services and in Haiti where those health workers perceived an initiative to reduce absenteeism favourably because they thought it promoted openness in the performance-based financing scheme [56]. On the other hand, lack of recognition for putting in effort dimmed motivation levels. Other motivators included a strong desire to help one's community and appreciation of the justification for a proposed QI project.

Motivation also arose out of observation of positive changes in the PHC setting due to QI and this was underscored by grateful clients or patients. Leadership by PHC facility and district QI mentors who remained committed and were able to showcase the use of context-specific data for QI was also found to enable QI. On the contrary, health workers did not like overlapping QI data streams because this, they perceived, stole time that they would otherwise spend caring for their patients.

Studies also reported the importance of buy-in by health workers and their managers into proposed QI interventions. This was signified by health workers embracing a spirit of personal sacrifice to receive public praise, including by PHC clients from the community. Further, research reports found that participants often embraced QI because they had grown dissatisfied with existing dismal quality of PHC services and felt an urgency to change [67, 74, 79, 81, 87].

Self-efficacy and capability to undertake QI was also highlighted in studies. A high level of technical and managerial proficiency acquired after implementing QI initiatives over time as reported from research in Kenya and Costa Rica [46, 48], promotes effective production, analysis, and use of PHC data for improvement. Moreover, participants in QI felt empowered and competent following training sessions which also served to help develop an understanding of their roles and responsibilities in QI [49] leading to increasing levels of comfort with QI approaches and methods [52]. Health workers reported that they could not spare time to attend QI meetings due to clinical engagements, a possible constraint. Other barriers reported in the literature included the sense of despair with which some easily gave up on QI initiatives when faced with multiple obstacles. An example of this came from a convergent mixed methods process evaluation of continuous quality improvement in South Africa [69] where health workers were discouraged by layers of managerial approval. In such cases across multiple PHC contexts, QI tasks were perceived to be time consuming—reducing health workers' confidence in the QI initiative—and abandoned [45, 48, 49, 52, 54, 58, 59, 62, 67, 69, 71–75, 81, 83–85, 90].

Health workers developed personal skills through their participation in QI initiatives. Skills such as empathy and enhanced communication with PHC clients reportedly led to deeper connections with fellow health workers but also clients. This was seen to facilitate QI. Still, familiarity with patient-centered approaches to PHC, regular review meetings where gaps and root causes to poor service quality were discussed, and internal supervision where knowledge was shared, and additional skills acquired was reported in the literature as important enablers. On the other hand, health workers in PHC who felt inadequately skilled in technical and clinical aspects and in the use of technology reported difficulties engaging effectively in QI [71, 74, 83].

Culture, comprised of shared norms, values, knowledge, artefacts, and practices, was found to play an important role in health workers' efforts to improve the quality of PHC. For example, QI efforts appeared to thrive in PHC settings with strong culture of using data to orchestrate healthcare improvements, where health workers' attitudes shift to focus more on the needs of patients (e.g., the desire to alleviate pain and reduce suffering), and where HCWs learn better and systematic approaches to solving problems [46, 47, 52, 55]. Additionally, culture of quality manifested in health workers being able to work across disciplinary boundaries,

where QI initiatives stir up healthy competition, and where participants reported collective responsibility for cohesion, meritocracy, a strong sense of taking responsibility for failure and success, and high standards in the PHC setting or workplace [55, 63]. Microsystem culture such as working with unsupportive colleagues where workload is not shared and characterized by a rejection of quality checklists [79] was found to be unsupportive of QI. In Indonesia, Limato and others [70] conducted 28 in-depth interviews in West Java Province. This led them to conclude that health workers at government-owned PHC facilities had a general tendency to reject transparency and accountability, which led to the failure of a QI initiative built around performance-based financing. Evidence on workplace culture's role in boosting or dooming QI interventions also came from other studies in multiple LMIC contexts [48, 54, 59, 60, 62, 64, 65, 68, 74, 81].

**Theme 2: Attributes of quality improvement intervention.** Component attributes of a QI intervention discussed under this theme include its strength and the quality of evidence underpinning it, how and whether participants perceive it to be beneficial, its cost, potential to be scaled up, and perceived sustainability. Other characteristics of the QI intervention that can enable or constrain its implementation may include its trialability (being trialed in small measures where potential failure is not catastrophic), the ease with which it can be integrated into existing health worker roles and tasks, and whether clients were afforded opportunity to shape its design. Rounding up the key attributes of any QI intervention is the source of the intervention which may dictate its acceptability, its complexity i.e., ease with which implementers understand it, scope, and disruptiveness during roll out; and closely related to this, feasibility (the extent that implementers feel confident that they can adopt the QI intervention) [26, 39].

SSA studies contributing to this theme included five each from Tanzania and South Africa; four from Ethiopia; and two each from Rwanda and Nigeria. Five African countries (Benin, Kenya, Mali, Namibia and Zimbabwe) each had a single-country study while Tanzania, Rwanda, Kenya, Nigeria Malawi, Burkina Faso, Mozambique, and Ghana were each part of a multi-country study. In Asia, Indonesia, Tajikistan, Sri Lanka and Papua New Guinea each contributed a study with India contributing two. Studies from Haiti and Costa Rica round up the list of those that contribute an understanding of enablers and barriers related to QI intervention attributes in PHC in LMICs.

QI implementation is enabled when health workers and managers perceive an intervention to be effective e.g., by observing the desired outcomes for patients and successful acquisition of new skills [41, 44, 62, 67, 72, 74, 75, 81]. A relative advantage accrues when implementers view a new QI initiative as better than current practice and when the intervention is designed to foster collaboration among a diverse team of workers, and even PHC clients. In contrast, QI is constrained when a QI project does not lead to any tangible improvement or is seen to bear negative or unanticipated consequences like creating an administrative burden for already overstretched HCWs that may manifests in multiple reporting channels. Other barriers were reported in the literature: an intervention package that does not envisage nor address other contextual and health systems barriers to successful implementation such as when was QI focused on short term technical fixes but did not address nor consider structural bottlenecks to PHC quality.

Cost, scalability, and sustainability aspects of QI were closely related. As enablers, the design of a QI intervention needs to make provision for long-term work to sustain changes while ensuring that its costs do not overwhelm the PHC system's capacity [48, 56, 62, 70]. At the same time, QI is scalable when QI interventions are perceived to be easily transferable to a new area of work within a PHC setting, to other health workers, or even to other health facilities by adopting small incremental changes rather than rapid disruptions [44, 52, 57, 72, 73]. Additionally, QI interventions are supported by health workers and health facilities when perceived

to be sustainable, i.e., when participants feel confident of continued implementation beyond the planned intervention period [45, 48, 50, 56, 62, 69, 72, 75, 80].

The significance of designing QI interventions in a manner that ensures that health workers see alignment between the proposed QI package and their everyday work responsibilities (job expectations in the PHC practice setting) while complementing participants' and health system's values was addressed by Ulrike Baker and colleagues [41] in their qualitative process evaluation of QI in Southern Tanzania and Mary Kinney and her counterparts [65] who used multiple case studies to understand sustainability of MPDSR in South Africa. Good examples of facilitating factors regarding trialability pointed to QI interventions that had been adapted and pre-tested to suit local conditions [42, 43]. Barriers that may thwart assimilation included new interventions that are difficult to integrate into routine PHC practice or those that require substantial modifications to service delivery workflows and an array of new skills for practitioners, new initiatives perceived to be inflexible or rigid, in addition to those that do not explicitly build on existing initiatives [44, 45, 78, 81, 86, 88, 90].

Paying attention to the preferences of PHC clients when designing QI interventions that affect them was thought to enable QI in addition to health workers' inputs to intervention design and was outlined by Mutambo and colleagues [77] who explored HCWs' perspectives during the setting up of child-friendly spaces in PHC clinics in KwaZulu-Natal, South Africa. However, Umunyana and others [71] in Rwanda and Olaniran and colleagues [86] in Nigeria reported that QI interventions that do not allow implementers to make or suggest adaptations might lead to such initiatives being viewed as alien and imposed, potentially leading to their rejection and failure.

Less complex QI interventions focus on a specific problem, are not too general and do not try to address too many things instantly or concurrently. These were some of the enabling factors identified in the literature. Other facilitating factors included having streamlined management structures in their design. Barriers identified by participants in relation to intervention complexity included those that are considered hard to understand, not easily translatable into tangible action plans, and QI interventions perceived as not user-friendly [42, 52, 66, 67, 70, 75, 81, 85, 86, 90], and found that QI projects considered feasible, timely and aligned local priorities were widely embraced, contributing to successful implementation.

**Theme 3: Organization and implementing team.** Buy-in (ownership), norms and culture, leadership, and decision-making at the organization level complement the tenure, prior experiences, cohesion, and skills of the implementing team to shape QI processes and outcomes. Also, maturity of the organization's approach to QI, presence of subject matter specialists able and willing to guide health workers at PHC facilities, and the participation of physicians in QI initiatives received important considerations in research reports included in this review and are described under this theme.

Studies outlined the barriers to and enablers of PHC QI at the meso level in 15 different countries in SSA as reported in 36 different articles. The SSA countries include Uganda, Rwanda, Ethiopia, Burkina Faso, Mozambique, and Mali that are low-income settings; lower middle-income countries of Benin, Ghana, Kenya, Malawi, Nigeria, Tanzania, and Zimbabwe; and South Africa and Namibia being upper middle-income settings.

Ensuring that leaders, managers, health workers and other stakeholders buy in to QI initiatives in PHC emerged strongly from the literature. Baker and others [41] found that health care workers (HCWs) were more receptive to continuous quality improvement (CQI) and welcomed on-job-training meant to bolster their skills in Southern Tanzania. This was echoed by Coulibaly and colleagues [81] in Mali where positive reception of a performance-based financing scheme for improving PHC services was noted among the initiative's strengths. Elsewhere, adequately preparing the team prior to introducing QI, having point persons to champion QI

in the health facility and PHC network, managers and team members who do not mind taking up additional or new responsibilities and an enthusiastic team that readily and publicly commit to PHC quality improvements were also important enablers of QI [46, 47, 50, 52]. In areas where there was little buy in, such as in Papua New Guinea [67] where regional managers exercising their hidden powers opposed QI, and in Indonesia [70] where 'ego programming', the tendency by those that perceive themselves to be outside a QI programme to decline participation, QI initiatives faltered. Organizations also rejected QI outright, with some declaring proposed interventions to be unsuitable without due consideration e.g., in Uganda [68] while middle managers in decentralized PHC settings simply went missing and did not cooperate or support frontline HCWs with QI efforts e.g., in Rwanda and Malawi [73].

QI interventions can flourish in organizations and teams with the right norms and where culture is supportive. A new way of solving intractable problems, regular team reviews that are focused on quality of care [52], finding ways to cope positively with scarcity when resources aren't adequate and lack of control at lower levels in centralized PHC settings [55] were mentioned. A quality culture with shared values, attitudes, practices at the organization level includes regular data analysis that drives action and improvement cycles, with feedback loops built around effective communication where QI progress is shared with stakeholders who in turn are responsive. Some downsides to quality culture reported in the literature include unchallenged absenteeism by HCWs [56]; decreasing concern for and normalization of common adverse PHC outcomes [59]; adversarial relationships between managers and HCWs; and a perversive lack of accountability where no follow up is done to ascertain achievement of agreed QI work plan targets [62, 76, 79], which constrain the ability of PHC to meet patient and client needs.

The maturity of an organization in undertaking QI was reportedly facilitated by accreditation processes which inspire a virtuous cycle of QI. Organizations undergoing accreditation are expected to plan for QI, allocate budgets and subsequently avail resources needed to enhance the quality of PHC services over time [67]. But the presence of concurrent and similar QI programmes in the same organization might introduce fragmentation and bring about confusion regarding organizations' priorities, a potential barrier [70]. Lack of institutional knowledge, where implementers do not fully understand organizational bureaucracies, can also hamper QI [80] where planned changes are complex and system wide. QI teams with short tenure due to high staff turnover appeared to reduce organizational maturity for QI implementation, e.g., in Benin where QI team members took up new jobs, and lack of community support and irregular monetary incentives affected teams' longevity [58].

Using pre-post designs with interviews and focus groups, the role of leadership in facilitating QI was reported by Limato and colleagues [70] in Indonesia and Nahimana and colleagues [84] in Rwanda where leaders owned and steered interventions. In contrast, Hounsou and colleagues [89] using mixed methods reported that a lack of interest by managers constrained MPDSR in Benin. Senior leaders, especially, need to actively embrace and publicly show support for QI for it to succeed as health workers do not wish to second guess their bosses' allegiances [43, 56, 64, 69, 77, 82]. While such champions can drive change within organizations and foster acceptance of QI initiatives, wearing too many hats can contribute to a lack of focus and become a distraction for QI. Weak leadership by governments in LMICs especially means that QI stewardship and monitoring was frequently left to donors and external partners, and this is in part because of lack of clarity in QI leadership arrangements and high turnover of leaders. In a sub-unit in Ethiopia, for example, leadership constantly changed hands [54]. Similarly, Eboreime and colleagues [45] linked weak leadership to organization culture unfavorable for QI, which proved detrimental to efforts to strengthen PHC quality in Kaduna state in Nigeria.

Physician involvement in QI also acted as an enabler and a barrier, depending on the context. Physicians assume leadership and help build other health workers' skills. However, in Karnataka State in India [49] found QI constrained in situations where the physician over-asserted authority and ignored other team members' contributions. Findings in Bihar [59], still in India, also highlighted the important gap left when doctors did not take up their roles as QI mentors in the context of management of birth complications for newborns, with fatal consequences.

Positive team experiences from successful legacy QI projects also reportedly produced domino effects e.g., in Tanzania [90] cross-pollination of ideas occurred when successful initiatives were shared across institutions. Incidentally, in both South Africa [65] and Southern India [49] strong social networks among health workers enabled QI whereas less cohesive teams reported worse outcomes. Strong teams also reported better, inclusive decision-making from the start of a QI project and balanced top-down and bottom-up approaches in decision making. Here, diversity was a strength as everyone was involved. A good example came from Uganda [68] where Hutchinson and colleagues used qualitative methods to study collaborative improvement (CI) for malaria surveillance. They report that CI was undertaken by small, committed teams who willingly involved patients and volunteers. Conversely, barriers to QI arise when team leaders do not genuinely involve others like non-technical (auxiliary) staff, who begin to feel sidelined.

Elaborating on the importance of subject matter specialists for advancing QI initiatives, in Uganda [50] participants received excellent support from a mental health specialist who had good knowledge of clinical practice guidelines, joining champions to bolster QI. The development of skills and knowledge also increases when trained team members report back to fellow health care workers, enabling key QI concepts such as Pareto charts, root cause analysis, and PDSA cycles to percolate in the team for a shared understanding [71, 72], with regular on-job training [69]. One-off training that leaves QI team members without adequate knowledge and skills needed to implement QI were characterized as barriers [72].

**Theme 4: Health systems support and capacity.**   Availability, adequacy, and distribution of resources needed to deliver PHC services to communities were key contextual drivers for QI reported in studies. Studies found weaknesses in PHC systems pillars required for quality enhancements, signifying inadequate capacity for QI. These include gaps in staffing, supplies and commodities, equipment and devices, physical space and infrastructure, data infrastructure and reporting, learning and knowledge systems, management of patient referrals, and leadership and governance. Some enablers of and barriers to QI under this theme e.g., those relating to leadership and management and to staff training and development, inevitably affect and are affected by those discussed in the other themes in this review. Tellingly, no country among the LMICs studied reported adequate or excess levels of resourcing for QI. Consequently, most of this section describes barriers to QI rather than enablers.

Forty-two studies highlighting various aspects of health systems support and capacity came from 13 different SSA countries. There were also five studies conducted in four Asian countries (India, Indonesia, Sri Lanka and Tajikistan) and two studies from Latin America (Haiti and Costa Rica).

The first barrier to QI in LMICs concerns a dearth of health workers which pervades health systems and within these, PHC delivery structures do not appear exempt. Low numbers, frequent leave of absence, and rapid turnover of staff are each associated with high workload and were reported as important constraints to QI [43, 50, 62, 63, 66, 68, 73, 76, 83, 84, 88, 89]. Where staff are available, aligning job descriptions and incentives appeared in the literature as a potential enabler of QI.

Adequate, well designed physical space aids intuitive flow of clients, encourages health workers to undertake certain tasks that are important for quality of care such as handwashing or waste segregation, or even providing oversight to acute cases in the newborn unit from the nurses' station. On the other hand, literature pointed to sub-optimal infrastructure (poorly designed) and or limited physical spaces as barring improvement actions [51, 77, 79, 81, 90]. This manifested as lack of much needed laboratories and pharmacy stores in Sri Lanka [76], for example.

Studies discussed the role of medical equipment and data infrastructure in relation to QI [69, 71, 75, 77, 81, 83, 85, 88]. Participatory and data-driven QI activities, revising data and tools to ensure harmonization of reporting systems were found to facilitate QI. Inadequate patient records at the PHC facility level as well as a lack of equipment, on the other hand, were mentioned as constraining attempts to enhance PHC service delivery and quality. As with equipment and staffing, stockouts of essential supplies and medicines was also reported as barrier to QI in PHC settings in LMICs including but not limited to Sri Lanka [76], India [78], Ethiopia [82], Nigeria and Tanzania [83, 86] and Rwanda [84].

Availability of resources to support QI was the focus of studies in Kenya [46], Uganda [52, 57], Democratic Republic of the Congo [51], Zimbabwe [53], Ethiopia [54, 72, 82], Malawi [55, 73], Haiti [56], India [59], Benin [58, 89], South Africa [60, 65, 69, 75], Tanzania [42, 61, 90], Zimbabwe [62], Rwanda [64], Namibia [66], Indonesia [70], Mozambique and Burkina Faso [79], Tajikistan [80], Mali [81], and Nigeria [44, 45, 86], underscoring its importance to impede QI and shared concerns across many LMIC contexts.

Studies in Rwanda [64, 71], in Namibia [66], in India [78], and in Mali [81] described the need for strong patient referral systems because continuity of care is integral to PHC. Inadequate patient referral systems, they reported, affected QI where the initiative aimed to enhance linkage and networking within a care network. Other enablers uncovered took the form of continuing (medical/health/nursing) education [47] and knowledge exchange platforms [48, 50, 54, 80]. Knowledge exchange platforms, it was reported, could enhance chances of successful QI by breaking down silos and fostering the integration of care packages.

As previously reported under microsystems and QI team and organization support, facilitative and regular follow up and mentorship enabled QI to happen in LMICs. Facilitating aspects such as feedback from the district health management team and mentorship for frontline HCWs supported skills-building and enabled implementers to brainstorm challenges. Unsurprisingly, QI implementing health workers found unpredictable follow up and punitive supervision geared towards fault-finding undesirable for efforts to improve the quality of PHC.

Quoting program and policy stakeholders in South Africa, Joan Mantell and colleagues [75] cite fragmentation in PHC as a key systems constraint for QI. Also, policies that limit access to PHC budgets as part of larger health systems configuration can also bar QI in LMICs. Conversely, Manisha Yapa and colleagues [69] report that availability of key guidelines and tools, and according to Werner et al. [80], national policies e.g., those that give a high visibility to PHC can indeed foster a supporting environment for PHC-focused QI.

Elsewhere, sub-optimal government policies and guidelines e.g., failure to integrate clinical decision support systems (CDSS) across the entire health system rather than in one or few vertical programmes was a key constraint contributing to non-use by trained health workers [74]. Mutambo and others [77] also observed that a government policy forbidding the clattering of walls had the unanticipated consequence of limiting the ability of QI implementers to decorate a children's clinic. The QI team had hoped to encourage play and boost service uptake by making the HIV clinic child friendly. Both studies were conducted in South Africa.

**Theme 5: External environment and structural factors.** The external environment forms the larger context in which QI interventions are implemented. It transcends the social,

economic, political, legal, and other normative aspects that shape societal and national health systems priorities and may indirectly or directly affect execution of QI projects or initiatives [26, 39]. In the present review, external incentives and societal pressures that drive change, macro-level allocation of resources and other externalities, and community characteristics such as social norms affect QI implementation in varied ways. Such structural factors are not enacted or imposed by social actors intending to shape QI interventions (although they may end up doing just that) but to address other intractable systemic or societal concerns. Thus, it is important for QI implementers, researchers, and policy makers to be aware of these and to make necessary adjustments to their QI programmes, where possible.

Evidence on external environment and structural barriers and enablers that affect QI implementation came from 19 countries reported in 31 studies. SSA contributed 26 studies from 14 countries while Asia had five countries' experiences reported in three studies. Six studies were conducted in Rwanda, five in Tanzania, four in South Africa, three in Ethiopia and two each in Malawi and Nigeria. Kenya, Burkina Faso, Mozambique, Namibia, Mali, Benin, Ghana, and Uganda in SSA and Papua New Guinea, Tajikistan, India, Indonesia, and Sri Lanka in Asia each had one research report included in this review.

Increased visibility of PHC business plans for donors, high level politicians and citizens in Tajikistan, and its high-level prioritization by the central government, was reported as an important enabler [80]. On the other hand, studies in Kaduna state in Nigeria [44, 45] reported that the government at state and national level had not prioritized PHC improvements and largely left the implementation of interventions geared towards PHC systems strengthening to donors, placing constraints on the relevant PHC Development Agency. Interestingly, weak coordination between the central government and semi-autonomous peripheral governments in Tajikistan thwarted the scale up of QI plans due to insufficient intergovernmental engagement [80].

Strong societal norms seep into the health system, through to individual health workers and managers, and shape contexts of health systems where QI is implemented. As an example, Hounsou and colleagues [89] used a retrospective mixed methods approach to explore implementation of the MPDSR mechanism in Benin and found that a culture of blame had a chilling effect in the reporting and audit of maternal deaths; a similar finding to Ayele et al. [63] in Ethiopia who also used mixed methods with administrative MPDSR data and in-depth interviews to report that health workers feared litigation and blame by relatives of deceased PHC clients. In this context, broad community dissatisfaction with explanations of causes of death and an overly litigatory society. However, in Mali, Coulibaly and colleagues [81] documented positive collaboration among health workers due to strong societal norms that encourage competitiveness, irrespective of place of employment. The inherent competitiveness inspired health workers to put in their best effort in QI implementation.

External pressures and incentives sometimes combined synergistically with socioeconomic policies to enable QI in PHC. This was the case in Tajikistan where the government introduced, rather serendipitously, a new health financing policy providing for per capita payments for PHC. The policy reduced financial barriers in the provision of PHC services. However, the QI research literature also reported areas where new policies had negative unintended consequences like the introduction of user fees in Rwanda which led to financial difficulties for women seeking ante-natal care, a component of a newly introduced QI package [84]. Expectedly, Wedernberg et al. [88] also reported socio economic challenges for patients that hindered access to PHC services in Rwanda.

Other external issues are more intractable. Impassable or unmotorable roads impede access to PHC clinics for communities and make it difficult for QI supervisors to undertake regular visits. Shaky internet constrains health workers' from downloading learning materials. And

extended power outages make life difficult for both managers and health workers alike. Good telephone connectivity may enable QI by making it easier for mentors to check in with front-line implementers without the necessity of a long, costly road travel. At the same time, good roads make travel within PHC networks easier for both communities and QI teams and supervisors. While responsibility for none of these structural issues lies within the health system, their inadequacies have the effect of introducing bottlenecks in QI efforts, especially in LMICs, where resources are scarce. Added to these, poor weather conditions, unsafe work environments, conflict, and security threats, further complicated matters, and may even see an exodus of skilled health workers besides diverting resources away from life-saving quality PHC. Expanding the list of challenges to QI that was found in the literature is the onset of COVID-19 pandemic which disrupted PHC in Sri Lanka, as was possibly the case globally in early 2020. Multiple research [44, 63, 67, 69, 76, 78, 79, 82, 83] reported these macro level barriers in one form or the other. Nahimana and colleagues [84] add to this long list of protracted constraints detailing how a prolonged drought and famine and the resulting refugee crisis, as happened in in eastern Burundi, rolled back progress in improving PHC in Kirehe district in Rwanda.

**Theme 6: Execution of quality improvement intervention.** No QI intervention is going to attain the desired objective unless implemented. Although this theme is being presented last, it is perhaps the most insightful, following this comprehensive synthesis of the evidence on barriers to and enablers of QI in PHC in LMICs. Execution includes elements of dosage and reach, and how the QI intervention is executed (with scope, quality, time, and cost) to achieve the intended results.

The twenty-two studies that underly this theme came from 17 countries. Eighteen of those studies originated from thirteen countries in SSA whereas four studies from Asia were derived from four different country contexts. Of the 17 countries in total, six are low-income countries, nine are lower middle-income countries and two are upper middle-income countries. South Africa and Rwanda each had four studies; Ethiopia, Benin, Malawi, and Tanzania each had two studies included and the rest (Mali, Namibia, Papua New Guinea, India, Malawi, Kenya, Mozambique, Burkina Faso, Sri Lanka, Tajikistan, Ghana, and Nigeria) were covered by a single research report.

Dosage (frequency and intensity) and reach (coverage) of QI interventions largely determine whether a QI change package is successful or not. Thus, reaching adequate numbers of implementers with knowledge and skills, whether by offering training sessions repeatedly or targeting and delivering them when most participants are available, were deemed important enablers [41, 69, 70]. Developing results oriented QI work plans and executing these in a participatory manner, ensuring periodic verification of whether a QI intervention is being implemented as planned, using feedback data from PHC facilities, and rolling out a QI package incrementally—where subsequent sessions build on earlier ones in a responsive manner—also facilitated QI [42, 71, 76, 77, 80, 81, 85, 87]. Contrary to these, keeping a limited focus of QI throughout its implementation, not unfurling all planned aspect of an intervention, and late roll of only a few aspects posed major hindrances, signaling a lack of fidelity to the specific QI's design and intent [45, 79, 89], and its potential failure. This could be attributed to the lack of clear implementation plans, overly ambitious QI work plans, and skewing QI implementation from original plans under pressure from funders, which exacerbate the challenges of QI implementation.

Already described earlier, supervision and mentorship were identified by the health workers among the biggest enablers of QI during the execution stage, according to Umunyana et al. [71]. Baker and colleagues [41] also reported positive impressions of health workers from being visited at their host health facility by mentors and supervisors. However, such visits

needed to be reflexive (questioning own stance, habits, values, attitudes) and reflective (learning from everyday experiences) to enable QI. In the case of tech-driven QI such as electronic integrated management of childhood illnesses (eIMCI), promptness with which implementation challenges were addressed also counted as an enabler for improved practice. Non-implementation of supportive supervision and limited training for implementers was identified as a constraint to QI [74]. When health workers do not practice new skills gained from QI for extended periods, they potentially forget QI techniques, underscoring the importance of ongoing support and mentorship [69, 76, 81]. Being humble and non-judgmental as a mentor-supervisor, Manzi and colleagues [87] reported, was preferred by PHC health workers following interviews and focus group discussion in Rwanda. Such mentors or supervisors assumed a wide range of roles such as facilitators, trainers, coaches, and role models [90] which enabled QI implementation. They could also act as champions, identifying blockers at various levels of the organization early enough and converting them to supporters thereby bolstering QI implementation [63, 66, 78, 88].

Engaging communities and targeting multiple stakeholders was further identified in research reports as key enablers, e.g., in Rwanda [64, 84, 88], Tanzania [42], Ethiopia [72, 82], India [78] and Nigeria [44] besides Burkina Faso, Malawi, Mozambique and Kenya [79], where QI implementers needed to work collaboratively with community resource persons and opinion leaders and make use of local knowledge to tailor their communication. As an enabler, engaging with a diverse array of QI stakeholders during implementation was specifically outlined by Kinney et al. [65] in South Africa, Basenero et al. [66] in Namibia, and Coulibaly et al. [81] in Mali. A boycott of QI by community catchments of PHC facilities happened in some instances where their local leaders had not been involved in QI implementation, constraining implementation. Also, QI activities geared towards improving access and quality of PHC services were hampered because clients kept off due to previous negative experience when seeking care, and because of limited risk communication by service providers. Nevertheless, reminders in home-based records for patients, where applicable, facilitated good communication between health workers and their clients [62, 63, 65, 66, 78, 81, 88, 90].

Another enabler during QI implementation entailed the redesign of work/patient flows, as described from stakeholders' experiences in South Africa, Rwanda, Uganda, Ethiopia, and Tanzania. Because sub-optimal physical infrastructure was identified as a key barrier to the provision of quality PHC, QI interventions that sought to re-design the clinic workflow, as needed, in a patient-centered manner, likely made it easier for health workers to adhere to care protocols.

Among others, [63, 86] found that QI implementation is more successful if it includes enhancements in documentation of care processes, and when stocks of key commodities are tracked and reported regularly. Conversely, failure by implementers to keep track of the availability of drugs and other stocks, aside from the actual stockout, constrains implementation. Further, QI roll out should pay due attention to limited staff time and competing tasks as described earlier, which can present significant challenges to participation by HCWs. Failure to consider this may mean that some staff miss numerous QI meetings and training sessions and place avoidable constraints on QI implementation [69].

## Discussion

This review aimed to identify the barriers (constraints) and enablers (facilitators) to QI in PHC settings of LMIC contexts. The review supports the notion, overall, that many contextual barriers exist that minimize the effectiveness of QI interventions, initiatives, or projects in these settings. At the same time, the review identified several factors that may promote the

implementation of QI interventions in this setting. Barriers and facilitators related to the inherent characteristics of the QI intervention, the immediate (micro) context, the implementing team and host organization at meso level, the larger health systems context, and at macro level, the societal and structural factors. Additional considerations are related to the execution of the QI intervention. These findings are important for those that design, promote, implement, regulate, and sponsor or fund QI. They are also important for users and clients of PHC services in LMIC countries because they point to how QI interventions can be further enhanced to support the attainment of PHC objectives of equitable, accessible, acceptable, timely, effective, and patient-centered care; and more broadly, health systems and societal development goals.

The results of this review, summarized by each theme, may help policy makers in many ways. First, they provide support for the design of integrated QI interventions as part of comprehensive (horizontal rather than vertical) approaches to QI, given the complexity of efforts required to successfully implement such interventions. Accordingly, considerations must be made for multiple intervening and interacting elements when formulating policies on quality of care to ensure that QI implementers are not caught off-guard or hamstrung by unforeseen (yet predictable) barriers. Secondly, the findings suggest that many QI interventions fail because the external environment and other structural forces constrain QI implementation despite the best efforts of implementing health workers. Systemwide investments appear to be more sustainable and scalable than small isolated experimental QI interventions that are largely donor dependent. The results also raise an intriguing question regarding the need to consider the perspectives of implementers (and PHC clients, where possible), rejecting top-down lift-and-shift QI initiatives. Surprisingly, the barriers and enablers are common, shared among multiple LMIC contexts with inherent transferable lessons. Given this, policy makers and gate keepers of PHC across LMIC settings may use the findings to call for more flexibility and enhanced financing, design, and execution of scaled up QI as part of accelerated investment in health systems towards UHC.

Reflecting on the review process, one of the challenges faced in selecting studies for inclusion concerned the definition of QI for which there is still no consensus. A second dilemma surfaced around the definition of PHC—and subsequent isolation of QI interventions in PHC—especially given the interconnectedness of PHC and tertiary (even secondary) care in any given health system. Consequently, decisions had to be made that both optimized sensitivity of the review and minimized selection bias, noting the lack of consensus, especially regarding the definition of QI. The review thus includes studies where actors at the micro, meso and macro levels actively sought to undertake quality QI for PHC using diverse approaches. Quality (healthcare) was broadly defined as that which is safe, effective, people-centred, timely, equitable, integrated, and efficient, following the WHO guidance [13]. WHO normative guidance plays an outsize role in the formulation of guidelines in LMIC health systems. Accordingly, QI was conceptualized as any deliberate intervention that aimed to enhance any, some or all these aspects of healthcare quality. The definition of PHC included clinical interventions of curative, rehabilitative and palliative nature, public health interventions meant to improve health at the population level including preventative interventions, and policy level interventions meant to affect health systems domains (financing, human resources, commodities and supplies, infrastructure etc.), if they targeted positive changes in health planning, resourcing, delivery, and outcomes at the district level and below. This inclusive, broad approach makes the review highly relevant to the diverse real-world LMIC contexts in which QI implementation takes place.

The systematic review, in analysing data from included studies, adopted MUSIQ model [26] and CFIR [39]. This review used concepts and categories from both frameworks to code and later organize the results thematically.

The review found that various barriers and enablers of QI in PHC in LMIC contexts relate to all the broad categories proposed by MUSIQ and CFIR frameworks, with many being inter-related, reflecting the complexity of health systems in which QI interventions are introduced, implemented, and thereby constrained or enabled. The Miscellaneous category under MUSIQ includes considerations related to the trigger for QI and whether QI tasks are strategic to the organization and were subsumed under the others in the present review.

Accordingly, MUSIQ model and CFIR proved useful for organizing the large amount of data derived from 50 diverse studies from equally varied countries and PHC settings. Additions to the CFIR framework [40] further helped with the synthesis and integration.

The results of this review echo those from an earlier umbrella review [25] which included reviews with primary research studies on the effectiveness, performance, and effects of quality management strategies in hospitals. They found 56 reviews focused almost exclusively on South-East Asia, Europe, and North America, with negligible research on the Americas and SSA contexts. Like this present review, Kringos and colleagues found that 35 of the 56 studies frequently reported contextual factors using the MUSIQ framework. The reported barriers and enablers included external environment, organization, QI support and capacity, microsystems, and QI team categories [25].

A more recent realist review [27] explored factors that affect the effectiveness of QI collaboratives (QICs), among the topics covered in the present review. Having synthesized the findings of 32 research abstracts, Zamboni et al [27] reported that factors inherent in external support, QI team, macro or structural aspects of implementation contexts can enable or constrain QICs, resonating with this review.

Like most previously published systematic reviews on QI that have tended to focus only on hospitals, Stokes and colleagues [22] synthesized research on barriers and enablers related to maternity care in LMICs. With a more limited database search covering only MEDLINE and CINAHL, they included nine studies, all of which were based on SSA. Seven of the studies reviewed by Stokes et al. [22] discussed clinical audits and feedback, like the five in this review that focused on MPDSR. A key finding of theirs, congruent with this present review, was that intrinsic motivation of health workers was a driver of the implementation of guidelines. However, the present review included community based PHC up to district hospital settings.

This review used an integrative approach [33] with results being synthesized narratively [34]. Studies were found on different topics including malaria surveillance, the application of digital technologies to improve health, expansion of access and quality of HIV/AIDS care, efforts to improve the quality of maternal, newborn and child health services, reduction of childbirth related deaths of women and newborns, and non-communicable diseases. Some studies were cross-cutting and did not look at specific packages of interventions within PHC. These were classified as either CQI or QIC if they explored QI processes using those two approaches, or PHC systems strengthening, if they were broad enough to include many domains of the health system. These categories helped to compare results and are not necessarily rigid or *a priori*. Importantly, there are many overlaps among them but this further demonstrates the suitability of the integrative and narrative approaches used for the review, given the review question.

Primary research studies also used many varied approaches to collect and analyze data on constraints and enabler of QI. Key informant interviews, in-depth interviews, semi-structured interviews, document reviews, field notes, participant- and non-participant observations, surveys, focus group and informal discussions, reflexive diaries, and health systems (administrative) performance monitoring were some of the data collection approaches used by QI researchers. Frequently, these were used in combination, with researchers aiming for data saturation. Sample sizes also ranged from a few tens to several hundred for both qualitative and

mixed methods design with homogenous and heterogenous groups of QI and PHC stakeholders. As well, included studies adopted a mix of varied frameworks including MUSIQ, CFIR, Tailored Implementation for Chronic Diseases (TICD), RE-AIM (reach, effectiveness, adoption, implementation, maintenance), COM-B (capability, opportunity, motivation, behaviour), PARIHS (promoting action on research implementation in health services), Breakthrough Model for Improvement, Positive deviance, Data to improvement pathway, and Adaptive management framework. These frameworks, where used, informed QI intervention design, data collection and analysis. Theories were also infused in QI research and included Force Field Analysis derived from Kurt Lewin's force field theory, Normalization Process Theory (NPT), Barth's Transactional Model of Culture, Gidden's Structuration Theory, and Carl May's Extended NPT. Perhaps given that most QI projects are part of implementation research initiatives, the extended use of frameworks and theories is not surprising. Due to the carefully thought-out theory-driven process evaluations, well-defined samples aiming for data saturation, and method mixing, studies were generally of good quality, having been subjected to critical appraisal, with congruent aims and methods, verifiable findings, and justified conclusions.

## Strength and limitations

The search for literature was comprehensive, covering all major health databases, grey literature repositories, selected websites, and even specialty journals. Moreover, no limiters were applied during search and retrieval. The selection of studies was guided by the review question and definitions adopted a broad and inclusive approach while guarding against scope creep—the tendency for reviews to balloon in size and become unmanageable. Studies were systematically screened and appraised for quality by two reviewers independently. Data extracted from 50 per cent of studies was compared between two independent reviewers for consistency. Together, these measures ensure that the review is relevant, with a low chance of bias, while being applicable across wide LMIC contexts. The review also included studies with a range of methods applicable to process evaluations that elicit contextual barriers to and enablers of QI initiatives in PHC. This was necessary to answer the review question comprehensively. Of note, the review found relatively recent articles and covered almost all countries in SSA, seven in Asia and two in Latin America, making it the most comprehensive of its kind so far. Lastly, the use of MUSIQ model and CFIR framework that are widely used in reviews and primary research on QI supported rigorous and transparent analysis.

Some limitations exist, nevertheless. Few studies were included from Latin America and Asia, the other continents with many LMIC countries. However, similarities in the contextual barriers to and enablers of QI in PHC irrespective of country context emerged during analysis, and are seemingly shared across LMICs in Africa, Asia, and Latin America. Still, policymakers and practitioners should carefully consider the contexts of included studies before transferring the review's conclusions to their unique PHC contexts. As there are ongoing debates regarding evolving definitions of QI, some researchers may avoid referring explicitly to QI, and such studies could have been missed. To mitigate this, a broad and inclusive definition that reflects the complex and interconnected nature of social, clinical, and public health interventions in the health system was applied to the review.

## Conclusion

This is the first review of its kind that synthesizes research on QI from LMICs with a focus on PHC. The uncovered themes related to barriers and enablers at the microsystem and individual health worker level, those intrinsic to the QI intervention, others that reside in the organization and team implementing QI, additional ones arising out of the larger health system,

external environment including the wider society, and how the QI intervention is executed. The review found many similarities and few contrasts among varied country contexts. Importantly, barriers and enablers are closely related and dynamic, likely affecting and affected by each other. The review found that relatively fewer (22) included studies exploring how the external environment and structural barriers and enablers affect QI implementation. It further found that how QI initiatives are executed had been explored in at least 17 countries out of all the six themes. This signals the opportunity for future research to investigate how wider (macro-level) issues and how the actual implementation process of QI is impeded or promoted to make PHC better for those that provide, use, fund, regulate or design it in LMIC contexts.

## Supporting information

**S1 Checklist. PRISMA checklist with preferred reporting items for the systematic review.**
(PDF)

**S1 Table. List of key words used for database search.**
(PDF)

**S2 Table. Studies grouped by geographic area and World Bank country income categories.**
(PDF)

**S3 Table. Topics covered in the QI research literature.**
(PDF)

**S4 Table. List of themes and sub-themes (barriers and enablers of PHC QI in LMICs).**
(PDF)

**S5 Table. Data extracts.**
(XLSX)

**S1 Fig. MEDLINE search strategy.**
(PDF)

**S2 Fig. ProQuest search strategy.**
(PDF)

## Acknowledgments

Professor Mark Limmer provided useful guidance in the development of the research protocol and during the conduct of the systematic review. Dr Tracy Epton of Manchester University also provided useful suggestions regarding the review methodology and approach, which shaped the conduct of this review. John Barbrook provided useful insights into the database search process, from a librarian information scientist perspective, which helped us to fine-tune our literature search strategy. Dr Kevin Oyula undertook quality appraisal and data extraction for one of the included research reports.

## Author Contributions

**Conceptualization:** Camlus Otieno Odhus.

**Data curation:** Camlus Otieno Odhus, Ruth Razanajafy Kapanga, Elizabeth Oele.

**Formal analysis:** Camlus Otieno Odhus, Elizabeth Oele.

**Methodology:** Camlus Otieno Odhus.

**Project administration:** Camlus Otieno Odhus.

**Software:** Camlus Otieno Odhus.

**Visualization:** Camlus Otieno Odhus.

**Writing – original draft:** Camlus Otieno Odhus.

**Writing – review & editing:** Camlus Otieno Odhus, Ruth Razanajafy Kapanga, Elizabeth Oele.

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
