## [Decision Letter · Decision Letter 0]

5 Oct 2023

PGPH-D-23-01492

Barriers to and enablers of quality improvement in primary health care in low- and middle-income countries: a systematic review

Dear Dr. Odhus,

Thank you for submitting your manuscript to PLOS Global Public Health. After careful consideration, we feel that it has merit but does not fully meet PLOS Global Public Health’s publication criteria as it currently stands. Therefore, we invite you to submit a revised version of the manuscript that addresses the points raised during the review process.

We look forward to receiving your revised manuscript.

Kind regards,

Loai Albarqouni, M.D. M.Sc. Ph.D.

Academic Editor

Journal Requirements:

1. Please provide separate figure files in .tif or .eps format only and remove any figures embedded in your manuscript file. Please also ensure all files are under our size limit of 10MB.

Additional Editor Comments (if provided):

Reviewers' comments:

Reviewer's Responses to Questions

**Comments to the Author**

1. Does this manuscript meet PLOS Global Public Health’s publication criteria? Is the manuscript technically sound, and do the data support the conclusions? The manuscript must describe methodologically and ethically rigorous research with conclusions that are appropriately drawn based on the data presented.

Reviewer #1: Yes

Reviewer #2: Yes

2. Has the statistical analysis been performed appropriately and rigorously?

Reviewer #1: N/A

Reviewer #2: N/A

3. Have the authors made all data underlying the findings in their manuscript fully available (please refer to the Data Availability Statement at the start of the manuscript PDF file)?

Reviewer #1: Yes

Reviewer #2: Yes

4. Is the manuscript presented in an intelligible fashion and written in standard English?

Reviewer #1: Yes

Reviewer #2: No

5. Review Comments to the Author

Reviewer #1: The authors have worked on a very important topic which will help to demystify the challenging situation which faces countries (at national, sub-national, and facility level) in sustaining quality improvement initiatives. As the world marks the half-way in the race towards achieving the Sustainable Development Goals (SDGs) 2030; especially in the SDG 3 in which quality of services is an integral component of the target 3.8 on Universal Health Coverage (UHC), this manuscript is very useful.

The paper is well conceptualized and written. However, there are minor areas which need some minor revisions (corrections/improvements) as explained below.

Title page

Key words: there is a repetition of the key word “primary health care”. Therefore, I suggest to the authors to delete one primary health care.

38 Introduction

Line 41: ADD the abbreviation LMICs in brackets as follows – countries (LMICs).

Line 47: replace “lower- and middle-income countries” with its abbreviation – LMICs.

Line 48: replace “lower- and middle-income countries (LMICs)” with its abbreviation – LMICs.

Line 58: replace “primary health care” with its abbreviation – PHC.

Line 62: replace “primary health care (PHC)” with its abbreviation – PHC.

Line 72: replace “quality improvement” with its abbreviation QI.

Lines 79-80: replace “quality improvement” with its abbreviation QI.

Lines 85-86: ADD the abbreviation for “sub-Saharan Africa in brackets so that it will read as follows ““……… therapy programmes in sub-Saharan Africa (SSA) [24].””

Line 95: replace “low- and middle-income countries” with its abbreviation LMICs

Lines 95-96: replace “quality improvement” with its abbreviation QI.

Line 96: replace “primary health care” with its abbreviation PHC.

97 Review aim and questions

Line 98: replace “quality improvement (QI)” with its abbreviation – QI.

Line 99: replace “primary health care” with its abbreviation PHC.

Replace “low- and middle- income countries (LMICs)” with its abbreviation LMICs.

Line 101: replace “primary health care” with its abbreviation PHC.

Lines 101-102: replace “low- and middle- income countries” with its abbreviation LMICs.

Line 104: replace “primary health care” with its abbreviation PHC.

Lines 106-107: replace primary health care” with its abbreviation PHC.

108 Review approach and methods

109 Review approach

Lines 123-124: replace “quality improvement” with its abbreviation QI.

Line 124: replace “primary health care” with its abbreviation PHC; and replace “low- and middle- income countries” with its abbreviation LMICs.

192 Data analysis

Line 193: replace “Quality Improvement (QI)” with its abbreviation QI.

Line 194: since the MUSIQ has already been written in long form in lines 87-88, I suggest to the authors to rephrase the sentence “The Model for Understanding Success in Quality (MUSIQ) developed by” to read as follows: The MUSIQ model developed by [27] was…….

Line 213: replace “primary health care” with its abbreviation PHC; and replace “quality improvement” with its abbreviation QI.

218 Results

219 Characteristics of included studies

Line 221: replace “primary health care” with its abbreviation PHC;

Line 222: replace “quality improvement” with its abbreviation QI.

Line 226: replace low- and middle-income countries (LMICs) with its abbreviation LMICs; and replace “quality improvement” with its abbreviation QI.

Line 227: replace “primary health care” with its abbreviation PHC.

Line 233: There seems to be an error OR a duplication of references. The references [41, 42] are not part of Table 2. Probably, the Authors meant References [71,72] which seem to be the same. Therefore, I suggest to the Authors to check on these references [41, 42] & [71, 74] both in the text and in the Reference List to ensure there is no any duplication.

275 Themes

Lines 276: replace “quality improvement” with its abbreviation QI; and replace “primary health care” with its abbreviation PHC.

Lines 277-278: replace the sentence “model for understanding success in quality (MUSIQ) and the consolidated framework for implementation research (CFIR),” with the following “"MUSIQ model and the CFIR"”

280 Theme 1: Microsystem and individual health worker(s) motivation

Line 291: replace “quality improvement” with its abbreviation QI.

Line 293: replace “primary health care” with its abbreviation PHC.

Line 299: DELETE “that” between workers & perceived.

Lines 341-342: replace “primary health care (PHC)” with its abbreviation PHC.

358 Theme 2: Attributes of quality improvement intervention

Line 359: replace “quality improvement” with its abbreviation QI.

Line 370: since “sub-Saharan Africa” has been mentioned in lines 84-85 with suggestion to ADD its abbreviation in brackets (SSA); and also, in line 227 & line 285 I suggested to REPLACE it with its abbreviation. Therefore, here (370) it is also recommended to replace “sub-Saharan Africa (SSA)” with its abbreviation SSA.

Line 377: replace “primary health care” with its abbreviation PHC.

Line 378: replace “Quality improvement” with its abbreviation QI.

Line 390: replace “quality improvement” with its abbreviation QI.

Line 403: replace “quality improvement” with its abbreviation QI.

Lines 404-405: since “maternal and perinatal death surveillance and response (MPDSR)” is well written and abbreviated in Table 2 in the row containing Ref. 62 (Kinney et al.[62]), I suggest to the Authors to replace “maternal and perinatal death surveillance and response (MPDSR)” with its abbreviation MPDSR.

428 Theme 3: Organization and 428 implementing team

Lines 432-433: replace “primary health care” with its abbreviation PHC.

Line 435: replace “primary health care” with its abbreviation PHC & quality improvement” with its abbreviation QI so that the line 435 will read as follows “Studies outlined the barriers to and enablers of PHC QI at”………

Line 436: replace “Sub-Saharan Africa (SSA)” with its abbreviation SSA.

Line 442: replace “primary health care” with its abbreviation PHC.

Line 448: replace “primary health care” with its abbreviation PHC.

Line 462: EDIT the word “arent” by ADDING an apostrophe between “n” & “t” as follows “aren’t” adequate……

Line 470: replace “primary health care” with its abbreviation PHC.

Line 472: replace “quality improvement (QI)” with its abbreviation QI.

526 Theme 4: Health systems support 526 and capacity

Lines 527-528: replace “primary health care (PHC)” with its abbreviation PHC.

Line 529: replace “primary health care (PHC)” with its abbreviation PHC.

Line 530: replace “quality improvement” with its abbreviation QI.

Line 540: replace “Sub-Saharan Africa” with its abbreviation SSA.

Line 544: replace “primary health care” with its abbreviation PHC.

Line 546: replace “quality improvement” with its abbreviation QI.

Lines 556-557: replace “quality improvement” with its abbreviation QI.

Line 559: replace “primary health care” with its abbreviation PHC.

Line 562: replace “primary health care” with its abbreviation PHC.

Lines 562-563: replace “low- and middle-income countries (LMICs)” with its abbreviation LMICs.

Line 565: replace “quality improvement” with its abbreviation QI.

Line 577: replace “quality improvement” with its abbreviation QI.

Lines 584-585: replace “primary health care” with its abbreviation PHC.

Line 587: replace “quality improvement” with its abbreviation QI.

Lines 591-592: replace “quality improvement” with its abbreviation QI.

601 Theme 5: External environment and structural factors

Line 602: replace “quality improvement” with its abbreviation QI.

Line 614: replace Sub-Saharan Africa (SSA)” with its abbreviation SSA.

Line 621: EDIT the word "reportedly" to be reported & INSERT “as" between “reported & an” so that it will read as follows: “““Tajikistan, and its high-level prioritization by the central government, was reported as an”””

Line 625: replace “Primary Health Care” with its abbreviation PHC.

Line 632: replace “maternal perinatal death surveillance and response (MPDSR)” with its abbreviation MPDSR.

Line 643: replace “quality improvement” with its abbreviation QI; & replace “primary health care” with its abbreviation PHC.

Line 659: replace “quality improvement” with its abbreviation QI.

671: Theme 6: Execution of quality improvement intervention

Line 672: replace “quality improvement” with its abbreviation QI.

Lines 674-675: replace “quality improvement” with its abbreviation QI.

Line 675: replace “primary health care” with its abbreviation PHC; and replace “low- and middle-income countries” with its abbreviation (LMICs).

Line 679: replace “Sub-Saharan Africa” with its abbreviation SSA.

Line 694: replace “quality improvement” with its abbreviation QI.

Line 709” EDIT the word “support” to be supportive so that the line will read as follows: ““Non-implementation of supportive supervision and limited training for implementers was””””.

Line 726: replace “primary health care” with its abbreviation PHC.

748 Discussion

Lines 749-750: replace “quality improvement” with its abbreviation QI.

Line 750: replace “primary health care” with its abbreviation PHC; and replace “low- and middle-income country” with the abbreviation LMIC.

Line 752: replace “quality improvement” with its abbreviation QI.

Line 754: replace “quality improvement” with its abbreviation QI.

Line 759: replace “quality improvement” with its abbreviation QI.

Line 760: replace “primary health care” with its abbreviation PHC.

Line 765: replace “quality improvement” with its abbreviation QI.

Line 766: replace “primary health care” with its abbreviation PHC.

Line 770: replace “quality improvement” with its abbreviation QI.

Line 772: replace “quality improvement” with its abbreviation QI; and replace “primary health care” with its abbreviation PHC.

Line 774: replace “World Health Organization (WHO)” with its abbreviation WHO.

Line 776: replace “quality improvement” with its abbreviation QI.

Line 778: replace “primary health care” with its abbreviation PHC.

Lines 785-786: replace “Model for Understanding Success in Quality, MUSIQ” with MUSIQ model [27]

Lines 786-787: DELETE “Consolidated Framework for Implementation Research” and KEEP only the abbreviation CFIR.

Line 789: replace “primary health care” with its abbreviation PHC.

Line 803: replace “sub-Saharan Africa (SSA)” with its abbreviation SSA.

Line 812: replace “quality improvement” with its abbreviation QI.

Lines 815-816: replace sub-Saharan Africa” with its abbreviation SSA.

Lines 817-818: replace “maternal and perinatal death surveillance and response (MPDSR)” with its abbreviation MPDSR.

Line 827: replace “continuous quality improvement” with its abbreviation CQI; & replace quality improvement collaboratives” with its abbreviation QICs.

Line 828: replace “primary health care” with its abbreviation PHC.

Lines 841-842: replace “quality improvement” with its abbreviation QI.

Line 842: replace “primary health care” with its abbreviation PHC.

857 Strength and limitations

Line 868: replace “quality improvement” with its abbreviation QI; and replace “primary health care” with its abbreviation PHC.

Line 870: replace “sub-Saharan Africa” with its abbreviation SSA.

884 Conclusion

Line 885: replace “quality improvement” with its abbreviation QI.

Lines 885-886: replace “low- and middle-income countries” with its abbreviation LMICs.

Line 886: replace “primary health care” with its abbreviation PHC.

Line 897” replace “primary health care” with its abbreviation PHC.

915 References

I suggest to the authors to re-check and ensure that REFs 41 & 42 are not duplicated in REFs 71 and 74.

Lines 1019-1025 (REF. 41 & 42):

1019 41. Baker U, Petro A, Marchant T, Peterson S, Manzi F, Bergström A, et al. Health workers'

1020 experiences of collaborative quality improvement for maternal and newborn care in rural Tanzanian

1021 health facilities: A process evaluation using the integrated 'Promoting Action on Research

1022 Implementation in Health Services' framework. PloS one. 2018;13(12):e0209092.

1023 42. Tancred T, Mandu R, Hanson C, Okuga M, Manzi F, Peterson S, et al. How people- centred health

1024 systems can reach the grassroots: experiences implementing community-level quality improvement in

1025 rural Tanzania and Uganda. Health policy and planning. 2018;33(1):e1-e13.

Lines 1111–1114 (REF 71) ; & Lines 1121-1123 (REF 74)

1111 71. Baker U, Petro A, Marchant T, Peterson S, Manzi F, Bergstrom A, et al. Health workers'

1112 experiences of collaborative quality improvement for maternal and newborn care in rural Tanzanian

1113 health facilities: A process evaluation using the integrated 'Promoting Action on Research

1114 Implementation in Health Services' framework. PLoS One. 2018;13(12):e0209092.

1121 74. Tancred T, Mandu R, Hanson C, Okuga M, Manzi F, Peterson S, et al. How people-centred health

1122 systems can reach the grassroots: experiences implementing community-level quality improvement in

1123 rural Tanzania and Uganda. Health Policy Plan. 2018;33(1):e1-e13.

Reviewer #2: The authors of this study conducted a systematic review of literature on the barriers and enablers of quality improvement in primary healthcare in low- and middle-income countries (LMICs). The study notion is interesting and it is well designed to answer the research objectives. However, the manuscript and some parts of the material need revision to make them publishable. My comments and suggestions in this regard are provided below.

1. General: the text throughout the manuscript needs a language and grammar edit by a professional.

2. Title and study design: although authors present this research output as a systematic review, it seems “scoping review” is a better title for this paper. I would like to refer the authors to a related publication about differences of these two types of review: https://link.springer.com/article/10.1186/s12874-018-0611-x, PMID: 30453902.

3. Style: please format the study style and organization and headings of manuscript based the journal requirements: https://journals.plos.org/globalpublichealth/s/submission-guidelines#loc-systematic-reviews-and-meta-analyses

4. Introduction: this section lacks an appropriate opening on the LMICs and why addressing the study notion in these countries is important.

5. Methods: the abstract mentions that four authors were involved in the process of title screening to data extraction and synthesis; however, the study has only three authors in this submission. Besides, reading the methods section thoroughly, shows different stages of review and data extraction were done by different authors and not by all of them. For example, the data extraction is done by one and half of the citations were re-checked by another author. As we know a rigorous systematic review needs incorporation of at least two independent reviewers screening the citations and extracting the data. Please revise the methods section in abstract and full text to clarify these points and clearly provide how many authors were involved in each stage of study.

6. Methods: the last part of this section entitled “Data analysis” might be better to be renamed to “Data analysis and synthesis”.

7. Results: regarding the six themes extracted from studies through the review, it is suggested to add a figure summarizing the key points of each theme so a brief summary of the findings could be reviewed by the audience of this study.

8. Discussion: it is highly suggested that authors included a paragraph about how the findings of this study may help health authorities and policy makers via the extracted themes to improve the quality of primary healthcare in LIMCs.

9. Supporting information and material could be appended into one file with a dedicated index page, for an easier access and use.

6. PLOS authors have the option to publish the peer review history of their article (what does this mean?). If published, this will include your full peer review and any attached files.

**Do you want your identity to be public for this peer review?** For information about this choice, including consent withdrawal, please see our Privacy Policy.

Reviewer #1: **Yes: **Eliudi Saria Eliakimu

Reviewer #2: **Yes: **Sina Azadnajafabad, MD, MPH

---

## [Decision Letter · Decision Letter 1]

7 Dec 2023

Barriers to and enablers of quality improvement in primary health care in low- and middle-income countries: a systematic review

PGPH-D-23-01492R1

Dear Mr. Odhus,

We are pleased to inform you that your manuscript 'Barriers to and enablers of quality improvement in primary health care in low- and middle-income countries: a systematic review' has been provisionally accepted for publication in PLOS Global Public Health.

Best regards,

Loai Albarqouni, M.D. M.Sc. Ph.D.

Academic Editor

Authors adequately responded to editorial and reviewers comments.

Reviewer Comments (if any, and for reference):

Reviewer's Responses to Questions

**Comments to the Author**

1. If the authors have adequately addressed your comments raised in a previous round of review and you feel that this manuscript is now acceptable for publication, you may indicate that here to bypass the “Comments to the Author” section, enter your conflict of interest statement in the “Confidential to Editor” section, and submit your "Accept" recommendation.

Reviewer #1: All comments have been addressed

Reviewer #2: All comments have been addressed

2. Does this manuscript meet PLOS Global Public Health’s publication criteria? Is the manuscript technically sound, and do the data support the conclusions? The manuscript must describe methodologically and ethically rigorous research with conclusions that are appropriately drawn based on the data presented.

Reviewer #1: Yes

Reviewer #2: Yes

3. Has the statistical analysis been performed appropriately and rigorously?

Reviewer #1: N/A

Reviewer #2: Yes

4. Have the authors made all data underlying the findings in their manuscript fully available (please refer to the Data Availability Statement at the start of the manuscript PDF file)?

Reviewer #1: Yes

Reviewer #2: Yes

5. Is the manuscript presented in an intelligible fashion and written in standard English?

Reviewer #1: Yes

Reviewer #2: Yes

6. Review Comments to the Author

Reviewer #1: (No Response)

Reviewer #2: Thanks for the revision. The manuscript is in a sound format. I have no further comments/suggestions.

7. PLOS authors have the option to publish the peer review history of their article (what does this mean?). If published, this will include your full peer review and any attached files.

**Do you want your identity to be public for this peer review?** For information about this choice, including consent withdrawal, please see our Privacy Policy.

Reviewer #1: **Yes: **Eliudi Saria Eliakimu

Reviewer #2: **Yes: **Sina Azadnajafabad, MD, MPH
